# MU-Bench: A Multitask Multimodal Benchmark for Machine Unlearning

**Jiali Cheng     Hadi Amiri**
University of Massachusetts Lowell
{jiali_cheng, hadi_amiri}@uml.edu

## Abstract

Recent advancements in Machine Unlearning (MU) have introduced solutions to selectively remove certain training samples, such as those with outdated or sensitive information, from trained models. Despite these advancements, evaluation of MU methods have been inconsistent, employing different trained models and architectures, and sample removal strategies, which hampers accurate comparison. In addition, prior MU approaches have mainly focused on *singular* tasks or modalities, which is not comprehensive. To address these limitations, we develop MU-Bench, the first comprehensive benchmark for MU that *(i) unifies the sets of deleted samples and trained models*, and *(ii) provides broad coverage of tasks and data modalities*, including previously unexplored domains such as speech and video classification. Our evaluation show that RANDLABEL (Graves et al., 2021) and SALUN (Fan et al., 2024b) are the most effective general MU approaches on MU-Bench, and BAD-T (Chundawat et al., 2023) and SCRUB (Kurmanji et al., 2023) are capable of achieving random performance on the deletion set. We analyze several under-investigated aspects of unlearning, including scalability, the impacts of parameter-efficient fine-tuning and curriculum learning, and susceptibility to dataset biases. MU-Bench provides an easy-to-use package that includes dataset splits, models, and implementations, together with a leader board to enable unified and scalable MU research.[1].

## 1 Introduction

Machine Unlearning (MU) aims at selectively removing a small portion of training data–and the influence of the samples–from a trained model. MU is essential for protecting sensitive information and discarding outdated samples. Recent works have studied machine unlearning in various contexts, including classification tasks on image (Guo et al., 2020; Tang et al., 2023) and graph (Chien et al., 2023; Cheng et al., 2023) data, multimodal tasks (Cheng & Amiri, 2023), generation tasks (Chen & Yang, 2023; Gandikota et al., 2023; Fan et al., 2024b), and federated learning (Wang et al., 2022).

Despite these advancements, existing approaches to machine unlearning face several challenges: (1): MU systems are evaluated under inconsistent settings, using different trained models (from which data is deleted) and metrics, which can lead to unfair comparisons and hinder the development of robust unlearning approaches (Fan et al., 2024b); (2): evaluation tend to focus on specific tasks, modalities, and architectures, which limits our understanding on the effectiveness of these models across different settings (Wang et al., 2023; Chundawat et al., 2023).

To address these limitations, we introduce MU-Bench, a comprehensive machine unlearning benchmark consisting of multiple tasks, data modalities, base models, standardized evaluation metrics, all compiled into an easy-to-use package with a leader board to enable robust and scalable MU research.

---

[1]Project page: `https://clu-uml.github.io/MU-Bench-Project-Page`.

38th Conference on Neural Information Processing Systems (NeurIPS 2024).

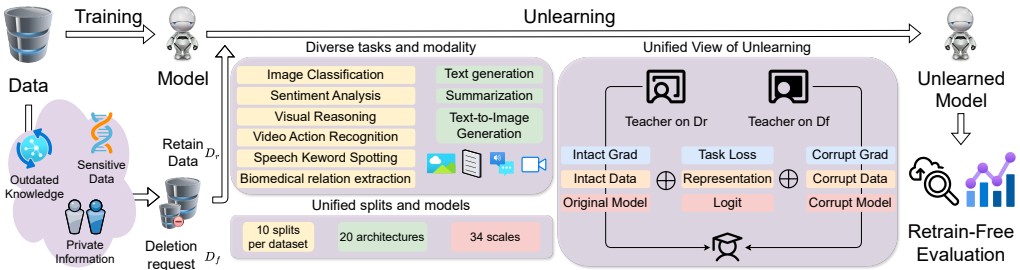

Figure 1: The MU-Bench benchmark for machine unlearning (MU) spans a comprehensive range of tasks and modalities, including previously unexplored data types such as audio, video, and biomedical data. The open-source package of MU-Bench provides standardized (unified) data splits, implements a suite of commonly-used MU methods and their design choices, enables fast experimentation and fair comparisons across MU methods, and is structured to easily incorporate new datasets and tasks in future.

To the best of our knowledge, this benchmark represents the first effort to benchmark existing MU approaches across a wide range of settings.

Our contributions are:

- constructing the first comprehensive MU benchmark with a wide coverage of tasks, domains, and modalities, including previously unexplored areas, such as speech and video processing, and biomedical applications, for systematic evaluation of unlearning algorithms;
- unifying (and perhaps democratizing) MU with uniformed deleted samples and a wide range of trained models and architectures to enable fair comparisons between MU methods;
- identifying design choices that explain performance variations across tasks and modalities;
- investigating several overlooked aspects of unlearning, such as deletion capacity, parameter-efficient fine-tuning (PEFT), and the impact of curriculum learning and dataset bias to inform future research directions.

Extensive experiments show that RANDLABEL (Graves et al., 2021), BAD-T (Chundawat et al., 2023), and SALUN (Fan et al., 2024b) are generally robust MU methods. When operating under a fixed training budget of compute (FLOS), RANDLABEL and SALUN outperform BAD-T. We find that existing MU methods benefit from PEFT but much less than other learning tasks, where below is 50% of the entire parameters, below which the model cannot be trained. Moreover, Curriculum Learning techniques can help models forget less and does not facilitate MU in most cases. In addition, performance variations across different tasks and modalities suggest that specific design choices within MU approaches significantly influence their effectiveness. In particular, certain tasks such as audio and video classification, are challenging for existing MU methods.

By design, MU-Bench is structured to incorporate new datasets and tasks, and we will continue to expand its resources in future.

## 2 MU-Bench

We outline the design of MU-Bench, covering tasks, datasets, models, and evaluation metrics.

### 2.1 Problem Formulation

**Machine unlearning** Let $D_{\text{Train}}$ denotes the training dataset, $D_f \subseteq D_{\text{Train}}$ the subset to be unlearned, and $D_r = D_{\text{Train}} \backslash D_f$ the remaining dataset post-unlearning. Given a model $f$ trained on $D_{\text{Train}}$, machine unlearning seeks to remove the influence of $D_f$ from $f$ without affecting the knowledge it gained from $D_r$, without retraining from scratch. We term $f$ as the original model and $f'$ as the model post-unlearning. A successful unlearned model $f'$ should be minimally impacted by $D_f$, while maintaining the performance of $f$ on the original downstream test set $D_{\text{Test}}$.

**Evaluation Metrics** Evaluating the efficacy of unlearning is crucial for identifying models that are more secure and retain no/less memory of deleted data. While previous studies have employed different metrics, we propose a set of metrics that do not require model retraining: performance

Table 1: Example datasets currently available in MU-Bench, covering a wide set of tasks and data modalities from different domains. $|D|$ denotes the size of training data. In MU-Bench, we set the deletion ratio to a maximum of 10% of $|D|$. Rows labeled with * indicate new tasks and data modalities introduced in MU-Bench for machine unlearning.

| Dataset | Task | Domain | Modality | $|\mathbf{D}|$ |
|---|---|---|---|---|
| **Discriminative Tasks** | | | | |
| CIFAR-100 (Krizhevsky, 2009) | Image classification | General | Image | 50K |
| IMDB (Maas et al., 2011) | Sentiment classification | Movie review | Text | 25K |
| * DDI-2013 (Segura-Bedmar et al., 2013) | Relation extraction | Biomedical | Text | 25K |
| NLVR$^2$ (Suhr et al., 2019) | Visual reasoning | General | Image-Image-Text | 62K |
| * Speech Commands (Warden, 2018) | Keyword spotting | Commands | Speech | 85K |
| * UCF101 (Soomro et al., 2012) | Action classification | General | Video | 9.3K |
| **Generative Tasks** | | | | |
| SAMSum (Gliwa et al., 2019) | Text summarization | Chat dialogue | Text | 14K |
| * BioFact (Min et al., 2023) | Text generation | Biography | Text | 183 |
| Tiny ImageNet (Le & Yang, 2015) | Text-to-Image generation | General | Image-Text | 20K |

on test set $D_{\text{Test}}$ ($\uparrow$), performance on deletion set $D_f$ ($\downarrow$), performance on remaining set $D_r$ ($\uparrow$), unlearning time ($\downarrow$), and success rate of membership inference attack ($\downarrow$).

**Toward a retrain-free evaluation**   Early works in machine unlearning research often considered the model retrained from scratch on $D_r$ as the *gold* standard for $f'$, which is now recognized as an inappropriate design choice due to several issues: **First**, evaluating $f'$ based solely on its closeness or similarity to the retrained model can lead to false negatives. This is because the parameters of $f'$ may fall onto different distributions than the retrained model, but still achieve competitive unlearning performance. On the other hand, the parameters of two models can match even with completely different training datasets (Lamproudis et al., 2022). **Second**, retrained models cannot guarantee the privacy of deleted data in practice, often maintaining undesired high performance on $D_f$, as demonstrated by previous work (Cheng et al., 2023). **Third**, obtaining a precise $D_r$ can be impractical in cases where the goal of unlearning is to remove toxic content (Zhang et al., 2023; Ilharco et al., 2023) or abstract concepts (Gandikota et al., 2023). Such abstract concepts may not correspond to identifiable data samples. **Finally**, retraining a model from scratch on $D_r$ can be impractical or even impossible due to confidentiality constraints, proprietary data concerns, or because the data may no longer be available. In addition, retraining is often expensive, especially for large datasets or complex tasks such as multimodal learning or large language models (LLMs). Based on the above shortcomings, we advocate for a retrain-free evaluation of unlearning systems, a method that is increasingly recognized in recent works (Chundawat et al., 2023).

## 2.2   Datasets and Tasks

We adopt nine publicly available datasets covering a diverse set of discriminative and generative tasks and data modalities. As Table 1 shows, the discriminative tasks include CIFAR-100 (Krizhevsky, 2009) for image classification, IMDB (Maas et al., 2011) for sentiment classification, DDI (Segura-Bedmar et al., 2013) for relation extraction in the biomedical domain, NLVR2 (Suhr et al., 2019) for visual reasoning, Speech Commands (Warden, 2018) for keyword spotting, and UCF101 (Soomro et al., 2012) for action classification. The generative tasks include SAMSum (Gliwa et al., 2019) for text summarization, Biography (adapted from Min et al. (2023), see below) for text generation, Tiny ImageNet (Le & Yang, 2015) for text-to-image generation.

We build a new dataset for evaluating machine unlearning in large language models (LLMs), focusing on the removal of personal information, as a common unlearning request. This is a crucial tasks because for example, on social media, user can choose to delete their accounts or privatize them, resulting in a critical and perhaps legal impetus for machine unlearning. The dataset contains factual descriptions of 183 celebrities, obtained from (Min et al., 2023), to enable machine unlearning of personal data from LLMs.

These datasets were chosen for their relevance to practical machine unlearning tasks, their variety, including both well-established and under-explored datasets, and their capacity to highlight differ-

ences between unlearning methods across diverse tasks and modalities (as they have non-saturated performance). This datasets allow for large scale and fair evaluation of unlearning methods, and addresses gaps in current research in several unexplored areas in machine unlearning.

## 2.3 Unified Unlearning

To address inconsistencies in the evaluation of MU approaches, we unify critical aspects such as *the choice and size of deleted samples* ($D_f$), and *the baseline model ($f$)* from which data is removed. This unification allows for meaningful comparison and democratizes access through open-source tools.

**Deleted Samples** For each dataset, we randomly sample 1-10% of the training data as $D_f$, with increments of 1% to covers both typical and extreme evaluation settings. This approach reflects typical and realistic settings where a small portion of data is deleted (Golatkar et al., 2020; Chundawat et al., 2023; Cheng et al., 2023), and challenges the limits of unlearning methods without fundamentally altering the data distribution, as would be the case with more extensive data removal.

**Original Model** For each dataset, we train a set of commonly-used models on different architectures and scales, from which $D_f$ is deleted, to allow for robust and relevant comparisons. We train a total of 20 architectures and 34 scales, such as ResNet (He et al., 2016) (18, 34, 50 layers), ViT (Dosovitskiy et al., 2021) (Small, Base, Large), Swin-Transformer (Liu et al., 2021) (Tiny, Small, Base), MobilNet V2 (Sandler et al., 2018) for image classification; and HuBERT (Hsu et al., 2021b) (Base, Large, X-Large), Whisper (Radford et al., 2023) (Tiny, Small, Base), Wav2Vec2.0 (Baevski et al., 2020b) (Base, Large) for the audio classification. Additional details are provided in Appendix A.3.

**Example Usage** We include the datasets, standardized data splits, evaluation scripts, and unlearning methods within an easy-to-use Python package and integrate them with commonly-used packages such as PyTorch (Paszke et al., 2019), Huggingface Transformers (Wolf et al., 2020), and Diffusers (von Platen et al., 2022), containing pre-trained diffusion models for image and speech data. Users can initiate an unlearning experiment with minimal adjustment to existing script. All original model checkpoints are released for standardized unlearning and fair comparisons. We also host and maintain a leaderboard to rank methods overall and on individual tasks and architectures. For example, to remove 5% of training data from a BERT-base model trained on IMDB using BAD-T (Chundawat et al., 2023), only a minimal script modification is required shown in code example 1. This setup simplifies the unlearning process and enables rapid comparison against methods and architectures.

**Taxonomy of Unlearning Techniques: A Teacher-Student Framework** To provide a deeper understanding of the design choices of existing MU approaches and their performance differences, we introduce a taxonomy based on a unified teacher-student framework. In this framework, the desired unlearned model $f'$ seeks to selectively discard specific knowledge from the original model $f$ under the guidance of a "teacher.' As shown in Table 2, the design choices of the teacher vary across different methods mainly from three aspects:

- **Knowledge Measurement (KM)**: the key question of how knowledge is quantified, which is determined by task loss (Loss), representation (Rep.), or output logits (Logit) in existing MU models;
- **Knowledge Corruption on $D_f$ (Corrupt)**: the key question of how the knowledge associated with $D_f$ is degraded, which is currently determined using techniques such as reversing gradients (NEGGRAD), using random data (RANDLABEL), or employing an incompetent teacher (BAD-T); and
- **Knowledge Retention on $D_r$ (Retain)**: the key question of how to preserve knowledge from $D_r$, which is typically achieved by treating the original model $f$ as the teacher.

These elements combine differently across methods, influencing both the teacher's role on $D_f$ and $D_r$, as detailed in Table 2; specifically, (i) and (ii) lead to teacher on $D_f$, and (ii) and (iii) lead to teacher on $D_r$. In Addition, the trainable parameters can be dense or sparse and internal or external. We utilize this taxonomy to categorize common and distinctive design elements in existing methods. This categorization helps in understanding how different unlearning approaches function and enables their transfer and adaptation to new contexts, such as generative tasks.

Table 2: Taxonomy of unlearning techniques. Despite different formulations and loss functions, existing approaches can be viewed in a unified teacher-student framework, with three design choices: (i) knowledge measurement (KM), (ii) knowledge corruption on $D_f$ (Corrupt), and (iii) knowledge retention on $D_r$ (Retain). The combination of (i) and (ii) leads to teacher on $D_f$, while combination of (i) and (iii) leads to teacher on $D_r$. For teachers on $D_f$ and $D_r$, Loss represents the expected task loss $\mathbb{E}_{(x,y)\in D} \sum L(f(x),y)$ on $D_f$ and $D_r$. Rep. denotes the KL Divergence of output distribution $\mathbb{E}_{(x,y)\in D} \sum \mathrm{KL}(f'(x), f(x))$ on $D_f$ and $D_r$. Trainable parameters are denoted as Dense or Sparse, and Internal or External.

| Method | Teacher on $D_f$ | | Teacher on $D_r$ | | Parameters |
| | KM | Corrupt | KM | Retain | |
|---|---|---|---|---|---|
| Exact unlearning | – | – | Loss | $f$ | Dense, Internal |
| NEGGRAD (Golatkar et al., 2020) | Loss | Grad | – | – | Dense, Internal |
| RANDLABEL (Graves et al., 2021) | Loss | Data | Loss | $f$ | Dense, Internal |
| BAD-T (Chundawat et al., 2023) | Logit | Model | Logit | $f$ | Dense, Internal |
| SCRUB (Kurmanji et al., 2023) | Loss | Grad | Loss + Rep. | $f$ | Dense, Internal |
| SALUN (Fan et al., 2024b) | Loss | Data | Loss | $f$ | Sparse, Internal |
| $l_1$-sparse FT (Jia et al., 2023) | – | – | Loss | $f$ | Sparse, Internal |
| MultiDelete (Cheng & Amiri, 2023) | Rep. | Data | Rep. | $f$ | Dense, Internal |
| EUL (Chen & Yang, 2023) | Loss + Rep. | Grad | Loss + Rep. | $f$ | Dense, External |
| UL (Jang et al., 2023) | Loss | Grad | – | – | Dense, Internal |
| GNNDELETE (Cheng et al., 2023) | Rep. | Data | Loss + Rep. | $f$ | Dense, External |
| SGA-TAU (Barbulescu & Triantafillou, 2024) | Loss | Grad | – | – | Sparse, Internal |

**Extension to generative tasks**  Even though many unlearning methods are designed for and evaluated on classification tasks, they can be applied to generative tasks with minimal modifications. For example, in case of RANDLABEL, data pairs $(x,y) \in D_f$ can be altered to $(x, y')$ where $y' \in D_r, y' \neq y$. For BAD-T, the method can be adjusted to match the predictions of each token when measuring the teacher-student divergence.

## 3 Experiments

**Setup**  For each dataset, we first train the task-specific original model $f$ long enough with hyperparameter optimization and select the best performing model. This is usually the practice for models deployed for real world applications. For LLM and Text-to-Image generation tasks, we evaluate unlearning from the pretrained models, since they are not fine-tuned for a specific task. In addition, we limit the unlearning time so that it does not exceed the retraining time, otherwise unlearning would not be practical. We repeat all experiments five times with different random seeds to account for stochastic effects. We focus on the following MU models selected based their widespread usage and unique characteristics: NEGGRAD (Golatkar et al., 2020), RANDLABEL (Graves et al., 2021), BAD-T (Chundawat et al., 2023), SCRUB (Kurmanji et al., 2023), and SALUN (Fan et al., 2024b). Details on the architectures used can be found in A.3 and the performance of the other MU models will be available on the leaderboard.

### 3.1 Main Results on Discriminative Tasks

As Figure 2 illustrates, NEGGRAD typically results in low performance on $D_f$, but severely compromises the knowledge on $D_{\mathrm{Test}}$ and $D_r$, indicating it is not an effective MU approach. In general, tasks like audio classification, video classification, text summarization and generation consistently challenge existing MU algorithms, potentially due to strong correlations within the data, see Figure 11-12). We report the average performance across all tasks as all metrics range from 0 to 100%.

For image classification on CIFAR-100, BAD-T achieves close-to-random performance on $D_f$ while preserving 40% accuracy on $D_{\mathrm{Test}}$ and $D_r$. Both RANDLABEL and SALUN effectively maintain models' capability on downstream test sets but fails to forget the deletion set. The original SALUN paper reported slightly different results, which we hypothesize may be due to the class-balanced sampling strategy and nuanced class hierarchy of CIFAR-100. Interestingly, SCRUB achieves very similar performances on $D_{\mathrm{Test}}$, $D_f$, and $D_r$, see Figure 7.

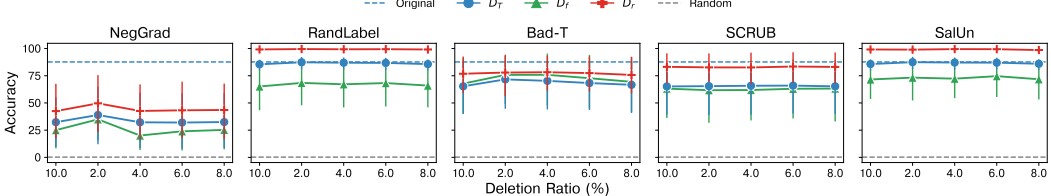

Figure 2: Overall average accuracy across all discriminative tasks.

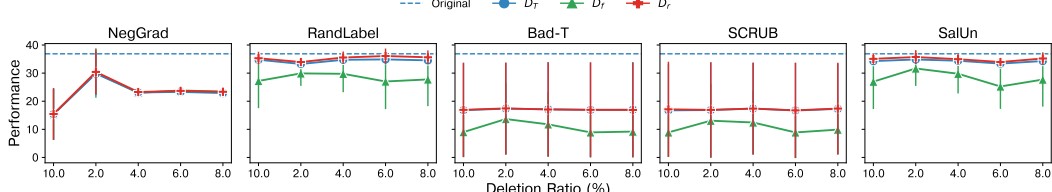

Figure 3: Overall average performance across all generative tasks.

For sentiment classification on IMDB, RANDLABEL and SALUN show promising results in forget $D_f$ with close-to-random performances, with minimal impact on $D_{\text{Test}}$. BAD-T and SCRUB also preserve strong performance on $D_{\text{Test}}$ but fail to unlearn $D_f$. Since IMDB contains strong dataset biases and shortcut features, corrupting the data labels implemented by RANDLABEL and SALUN seems to be a more effective approach than corrupting gradient, see Figure 8.

For biomedical relation extraction on DDI, RANDLABEL, SCRUB, SALUN all succeed in forgetting the deletion set $D_f$, with SCRUB slightly impairing test performance more than others. Conversely, BAD-T completely failed to unlearn $D_f$, see Figure 9.

For visual reasoning on NLVR2, RANDLABEL and SALUN again are successful in unlearning $D_f$, unlike BAD-T and SCRUB, which failed to unlearn $D_f$. However, one potential issue with SALUN is in the excessively low performance, almost close to zero performance, on $D_f$, which may be too low and prone to information leakage. This will be further discussed in §4, see Figure 10.

The speech keyword spotting on Speech Commands show that none of the existing methods can forget $D_f$ without severely impacting knowledge retention. Either with minimal knowledge removed (NEGGRAD, RANDLABEL, SALUN), or resulting in too much performance degradation on $D_{\text{Test}}$ (BAD-T, SCRUB). This can potentially be due to the correlations between audio waves, for which prior approaches do not have mechanisms to handle, see Figure 11.

For video action recognition on UCF101, all methods maintain original performance on $D_{\text{Test}}$ and $D_r$, but all fail to forget $D_f$, with 90+% accuracy. This can be attributed to the fact that current video classification methods rely on inter-frame correlation, while existing MU methods lacks mechanisms to remove such information, leading to failed unlearning, see Figure 12.

## 3.2 Main Results on Generative Tasks

In general, generation tasks present greater challenges for unlearning and evaluation. As Figure 3 shows, for text summarization on SAMSum and text generation on BioFact, existing general MU approaches all fail to achieve unlearning. RANDLABEL and SALUN has limited influence over all data including $D_f$ and $D_r$, while BAD-T and SCRUB remove knowledge of all data. In addition, we find that NEGGRAD show very different performance pattern on generative tasks compared to discriminative tasks, with non-random performance when a small portion of examples are deleted, see Figure 13.

For text-to-image generation, we find all methods can effectively reduce the clip score between image-prompt pairs on $D_f$ with limited impact on $D_{\text{Test}}$ and $D_r$, see Figure 15). To ensure the generated images are not from the orginal classes, we use a trained image classifier to classify the samples in $D_f$. SALUN outperforms all other approaches by 5.1 in accuracy on average, see Table 4.

Additional results on training time and membership inference attack are shown in Appendix A.5.

# 4 Discussion and Analysis

**What is the deletion capacity of each method?**     We define *deletion capacity* as the amount of data a model can forget without degrading performance on $D_T$. RANDLABEL and SALUN have relatively larger deletion capacity than SCRUB, while BAD-T has the smallest capacity. These results suggest that task loss is a potentially better way of knowledge measurement than matching logits in BAD-T. Another reason is the computation cost of BAD-T restricts its capability of forgetting more samples. Furthermore, we find that the deletion capacity of the same MU method varies across different tasks, modalities, and network architectures. SALUN has large deletion capacity on image and text classification datasets, but much smaller capacity on multimodal tasks, shown in Figures 7–10.

**Does unlearning amplify biases?**     A less explored aspect of unlearning in existing works is does MU amplify or restrict the model's dependence on biases in MU. To answer these questions, we evaluate the zero-shot transfer performance of $f$ and $f'$ on test examples that are adversarial or from shifted distributions, specifically, CIFAR100-C (Hendrycks & Dietterich, 2019) for CIFAR100, Rotten Tomatoes (Pang & Lee, 2005) for IMDB, extra test set from (Suhr et al., 2019) for NLVR2, UCF101-DS (Schiappa et al., 2023) for UCF101, and XSum (Narayan et al., 2018) for SAMSum. The results show that NEGGRAD significantly affects models' capability on transfer test sets, while other methods we evaluated do not strongly influence models' dependence on biases, see Figure 4.

**Does unlearning follow scaling laws?**     Scaling is a critical aspect to understand the limitations of an unlearning method. The results show that RANDLABEL, SALUN, NEGGRAD, and BAD-T have a better predictability of performance on $D_f$, given the amount of compute (FLOS), while the performance of SCRUB depends on the switch between max steps and min steps. In addition, NEGGRAD and SCRUB have faster speed in decreasing performance on $D_f$. BAD-T has relatively slower speed, due to the fact that it simultaneously iterate through $D_f$ and $D_r$ at every optimization step, which leads to more computing cost than other methods.

**Does unlearning benefit from curriculum learning?**     The effect of curriculum learning (Bengio et al., 2009; Sukhbaatar et al., 2018) (CL) in MU is an overlooked aspect in existing literature. MU models often sample batches randomly with no specific order and treat inputs with equal weight. We experiment with one common curriculum learning approach SuperLoss (Castells et al., 2020), which implements the core principle of curriculum learning. Specifically, it weights training losses based on sample difficulty, weighing down the contribution of samples with large training loss (potentially hard examples) to allow the model to learn from easier samples. As through training, the loss of the hard examples decreases, hard examples are gradually introduced for training. The results show that overall SuperLoss results in a slightly larger performance on $D_f$, indicating CL is likely to help model forget less. One exception is that on Speech Commands, CL outperforms Non-CL by 25.4 in accuracy. We defer further experiments with other CL techniques to future work.

**Does unlearning benefit from parameter-efficient fine-tuning (PEFT)?**     Despite recent advancements of parameter-efficient fine-tuning (He et al., 2022), most MU methods optimize the entire network parameters, which results in significant cost. Only a few approaches have adopted a parameter-efficient strategy (Chen & Yang, 2023; Cheng et al., 2023). Since PEFT only updates a small portion of the model, it is intuitive to assume that PEFT can maximally retain the knowledge from the original model without compromising unlearning. To validate this hypothesis in the context of MU, we experiment with LoRA (Hu et al., 2022a). The results show that most methods can benefit from PEFT, where the performance gap on $D_f$ is less than 10 points in accuracy. However, the amount of trainable parameters in MU is much larger than that of fine-tuning. As the trainable parameters are less than 50% of the original size, the performance on $D_f$ is close to that of $D_r$. Such performance persists even with larger learning rate and longer training time, indicating unlearning $D_f$ cannot be achieved below the threshold of 50%, see Figure 6. This minimum trainable threshold (Hu et al., 2022b; Su et al., 2023) is much larger than non-MU tasks with as low as a few thousand parameters, since selective knowledge removal is a more challenging task. Meanwhile, the performances on $D_{\text{Test}}$ and $D_r$ are not affected, indicating LoRA forgets less and slower in MU.

**Which design choices are effective for machine unlearning?**     For discriminitive tasks, corrupting gradient is a less effective approach compared to corrupting data (RANDLABEL, etc.) and model (BAD-T). Corrupting gradients can discard learned knowledge and therefore we suggest not using

it in isolation without other constraints. However, this approach has a greater potential for generative tasks. It is generally more effective to simultaneously iterate through $D_f$ and $D_r$ (BAD-T) or randomly iterate through the training set (RANDLABEL, SALUN), than to clearly separate $D_f$ and $D_r$. For example, SCRUB takes a few passes on $D_f$ to forget the deletion set before learning on $D_r$ to retain non-deleted data. On the other hand, simultaneous processing of $D_f$ and $D_r$ lead to higher computational cost.Using representation or task loss as a measurement of learned knowledge can adapt to both discriminative and generative tasks, while using logits (BAD-T) has a much more restricted application merely on classification tasks.

**Is a lower performance on $D_f$ always better?** Previous works focus on driving the performance on $D_f$ to as low as possible. We suggest that excessively low score on $D_f$ might reveal information or indicate its existence, which may be taken advantaged by adversary. Moreover, unlearning does not mean a model should completely lose its capability of handling specific samples in $D_f$. Instead, a balanced approach where the unlearned model maintains a reasonably low performance on $D_f$ is preferable. Recent works are focusing on this direction, such as zero-retrain evaluation (Chundawat et al., 2023), knowledge gap on $D_f | D_T$ (Wang et al., 2023; Cheng et al., 2023). We defer further analysis on the desirable performance of $D_f$ to future work.

## 5  Conclusion

**Conclusion**  We propose MU-Bench, the first comprehensive machine unlearning (MU) benchmark that spans various tasks and data modalities, including those previously unexplored in MU. We introduce a unified taxonomy of existing MU works, which highlights their unique design choices and establishes connections between them. We also conduct extensive experiments with commonly-used and recent MU algorithms using MU-Bench, discovering that audio and video tasks require more focused development of MU techniques. In addition, we explore several overlooked yet crucial aspects of unlearning, such as bias, parameter-efficiency, curriculum learning, and deletion capacity. Finally, we develop an open-source package of MU-Bench to provide unified data splits, and implement a suite of commonly-used MU methods and their design choices to enable fast experimentation and fair comparisons across MU methods. The package along with a leaderboard are structured to easily incorporate new datasets and tasks in future. We will continue to expand MU-Bench by incorporating more datasets and tasks.

**Future Works**  There are several venues for future work including: *(a): MU methods for under-investigated tasks and modalities*: existing unlearning methods are primarily developed for text or image data types. Our experiments on MU-Bench show that current models severely underperform in audio and video contexts. A promising area of research is to extend MU to these data modalities and tasks through focused development of MU techniques to ensures comprehensive MU capability. *(b) Efficient MU methods*: existing unlearning methods require extensive training, either tuning the entire model or training on large portions of the dataset. Meanwhile, most methods do not benefit from PEFT. Future research can focus on developing more efficient MU methods using approaches like zero-shot methods, sparse methods, and curriculum learning methods to speed up the unlearning process. *(c) Explainability*: understanding why certain samples are more easily forgotten than others could shed light on inner working of MU methods and improve MU performance. Therefore, investigating the complexities of samples that affect their retention or deletion is a promising area of research. *(d) Evaluation*: current evaluation of MU is still in its early stage and demands more attention. Refining current evaluation strategies and metrics will be crucial for advancing the field. *(e) Theoretical guarantee of MU*: most current non-DP-based MU approaches do not provide theoretical guarantees. A critical future directions is to develop theoretical frameworks that provide bounds performance bounds for MU.

**Limitations**  While our work marks significant progress, it has the following limitations: *(a): Not all MU algorithms are evaluated*: due to the significant cost and resource constraints, we focused on a selection of recent, well-performing and representative approaches rather than an exhaustive examination of all MU models. *(b): Breadths of experiments*. Our investigation into parameter-efficient fine-tuning and curriculum learning were limited to specific methods like LoRA and SuperLoss, though other more effective approaches exist. *(c): Not all tasks are included.* There are some relevant tasks that are not currently included in MU-Bench, such as those related to graphs, recommendation, or retrieval tasks. We plan to expand the range of tasks and datasets in ongoing development of MU-Bench.

## Broader Impact Statement

Our work lays a foundation for fair and consistent evaluation of machine unlearning techniques and its applications, including the Right To Be Forgotten (RTBF) in AI models, which ensures the protection of personal data and the integrity of AI systems.

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

# A    Appendix

## A.1    Related work

**Categorization of unlearning methods**    *Exact unlearning methods* divide the remaining data into several shards and train a separate model on each subset of data. Then all models are combined to make a prediction. They work under different scenarios, including on images (Bourtoule et al., 2021; Wu et al., 2020b,a; Liu et al., 2022b; Dukler et al., 2023; Lin et al., 2023), on graphs (Chen et al., 2021). *Differential Privacy-based methods* adopt a one-shot weight update followed by added noise to model weights, whose probability distribution is indistinguishable from that of a model retrained from scratch with theoretical guarantee Golatkar et al. (2020); Guo et al. (2020); Neel et al. (2021); Brophy & Lowd (2021); Wu et al. (2023); Izzo et al. (2021); Suriyakumar & Wilson (2022); Liu et al. (2023a). *Teacher-student unlearning methods* formulates unlearning as selectively transferring the knowledge into the unlearned model (student). Usually, the teacher on the non-deleted data is the original model, while the teacher on deleted data is opposite to the original model Wang et al. (2023); Kurmanji et al. (2023); Chundawat et al. (2023); Cheng et al. (2023); Fan et al. (2024b); Tarun et al. (2023a).

**Unlearning for discriminative tasks**    Unlearning works in discriminative tasks covers image classification (Foster et al., 2023; Lin et al., 2023; Jia et al., 2023; Zhang et al., 2022b), text classification Li & Liu (2023); Mehta et al. (2022); Cha et al. (2024); Kang et al. (2024), node / edge classification on graph-structured data (Chen et al., 2022b; Chien et al., 2023; Cong & Mahdavi, 2023; Cheng et al., 2023; Wu et al., 2023; Cheng et al., 2023; Sinha et al., 2023), regression (Tarun et al., 2023b), image retrieval (Zhang et al., 2022a), multimodal classification tasks (Cheng & Amiri, 2023; Poppi et al., 2024; Li et al., 2024b), Bayesian models Nguyen et al. (2020), recommender systems Chen et al. (2022a); Li et al. (2022, 2023), k-means (Pan et al., 2023), and intelligent agents (Liu et al., 2022a). Many other works focus on class unlearning, i.e. removing all samples with a specific class (Chen et al., 2023a). However, discriminative tasks on audio and video have been limitedly studied, which this work bridge the gap.

**Unlearning for generative tasks**   Unlearning for generation models centers on removing copyrighted, private, NSFW, or biased content from generative models, including diffusion models (Gandikota et al., 2023; **?**, 2024; Liu et al., 2024; Fuchi & Takagi, 2024; Fan et al., 2024b), image-to-image models (Li et al., 2024a), text summarization models (Chen & Yang, 2023), translation models (Wang et al., 2023), and text generation models (Lu et al., 2022; Jang et al., 2023; Kassem et al., 2023; Chen & Yang, 2023).

**Unlearning in LLMs**   Recently, more attention has been paid to unlearning in LLMs. Most works focus on gradient ascent to forget copyrighted content (Eldan & Russinovich, 2023). Yao et al. (2024) designed two additional losses: 1) predicting if answer is gramatically correct, and 2) maintaining performance. SOUL (Jia et al., 2024) leverages second-order optimization techniques. Other approaches include sparsity Ma et al. (2023) and operations on gradient Ullah et al. (2021); Hoang et al. (2024). Applications of unlearning include removing bias Setlur et al. (2022); Chen et al. (2023b), alleviating backdoor attack Wei et al. (2023), conducting data poinson attack Di et al. (2023).

**Unlearning evaluation**   Evaluation of MU include the effectiveness of exact / DP-based unlearning (Thudi et al., 2022), adversarially trained models Liu et al. (2023b), adversarially evaluation (Goel et al., 2022), red-teaming tool for concept removal Tsai et al. (2024), verification Sommer et al. (2022), sequential deletion (Gupta et al., 2021), vulnerability to attack ZHAO et al. (2023), trade-off with reverting decisions Pawelczyk et al. (2023), different choices of deleted points (Fan et al., 2024a), theoretical capacity of deletion (Liu et al., 2023a), under shallow models Schelter et al. (2021); Ginart et al. (2019), under zero-shot setting Chundawat et al. (2022).

**Task-specific MU benchmarks**   In general, datasets and benchmarks for unlearning is underexplored. Most works draw samples as deleted data from existing datasets and choose different subsets from paper to paper. UnlearnCanvas is a benchmark for unlearning for diffusion models (Zhang et al., 2024). TOFU (Maini et al., 2024) is a benchmark for unlearning fictitious author profiles in LLMs. Conversely, we test LLMs with unlearning real profiles, as such information appears in the pretraining corpus of the LLMs, which aligns with the unlearning setting.

## A.2   Implementation details

For all methods, we adopt a batch size of 32 and Adam optimizer. We search for the best learning rate in $[1e-5, 5e-5, 1e-4, 5e-4]$. All experiments are conducted on NVIDIA A100 GPUs.

## A.3   Original models

We release the following 20 network architectures and 34 different scales to serve as original models in our benchmark.

For CIFAR-100, we train ResNet (He et al., 2016) (18, 34, 50 layers), MobileNet V2 (Sandler et al., 2018), ConvNext (Liu et al., 2022c), ViT (Dosovitskiy et al., 2021) (Base, Large), and Swin-Transformer (Tiny, Base). For IMDB, we train BERT (Devlin et al., 2019) (base and large), DistilBERT (Sanh et al., 2020), and Electra (Clark et al., 2020) (Base). For DDI, we train BioBERT (Lee et al., 2019), PubMedBERT (Gu et al., 2021) (abstract only and full text). For NLVR2, we directly take the Vilt (Kim et al., 2021) model finetuned on NLVR2 from the original paper. For Speech Commands, we train HuBERT (Hsu et al., 2021a) (Base, Large, X-Large), Wav2Vec2.0 (Baevski et al., 2020a) (Base, Large), Whisper (Radford et al., 2022) (Tiny, Base). For UCF101, we train VideoMAE (Tong et al., 2022) (Base, Large). For SAMSum, we train T5-V1.1 (Lester et al., 2021) (Small, Base, Large, X-Large). For Biography, we directly take the instruction tuned Alpaca (Taori et al., 2023) (7B, 13B), Vicuna V1.3 (Zheng et al., 2023) (7B, 13B). For Tiny ImageNet, we directly take the Stable Diffusion V1.4 (Rombach et al., 2022) from the original paper.

```
from transformers import TrainingArguments, AutoTokenizer, AutoModelForSequenceClassification

# Additional code for unlearning
from benchmark import UnlearningTrainer, UnlearningArguments
unlearn_config = UnlearningArguments(
    unlearn_method="bad_teaching",  # MU method
    backbone="bert-base",           # Network architecture
    data_name="imdb",               # Dataset
    del_ratio=5                     # Standardized splits
```

Table 3: Contribution of curriculum learning in MU.

| Dataset | CIFAR-100 | IMDB | DDI-2013 | NLVR[2] | Speech Commands | UCF101 | SAMSum | BioFact | Tiny ImageNet | Ave |
|---------|-----------|------|----------|---------|-----------------|--------|--------|---------|---------------|-----|
| **Non-CL** | 55.5 | 68.6 | 53.4 | 58.1 | 42.8 | 76.6 | 28.5 | 17.4 | 21.1 | 46.9 |
| **CL** | 56.4 | 68.6 | 61.9 | 58.2 | 17.4 | 77.0 | 19.2 | 17.3 | 20.9 | 44.1 |
| **Gap** | -0.9 | 0 | -8.5 | -0.1 | 25.4 | -0.4 | 9.3 | 0.1 | 0.2 | 2.8 |

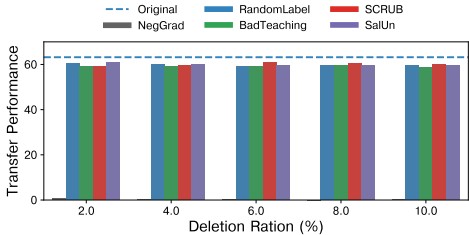

Figure 4: Transfer performances.

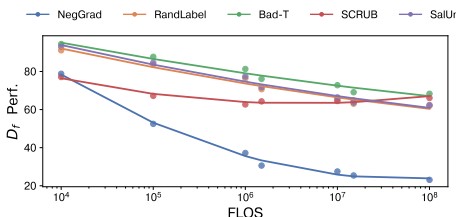

Figure 5: Scaling of $D_f$ performance.

```
)                                                                              10
                                                                               11
# Standard HuggingFace code                                                    12
args = TrainingArguments(output_dir="tmp")                                     13
original_model_name = f"mu-bench/{unlearn_config.backbone}-{unlearn_config.data_name}"  14
raw_datasets = load_dataset(unlearn_config.data_name)                          15
tokenizer = AutoTokenizer.from_pretrained(original_model_name)                 16
original_model = AutoModelForSequenceClassification.from_pretrained(original_model_name)  17
                                                                               18
# Replace original HF trainer with our new trainer                            19
trainer = UnlearningTrainer(model=original_model, args=args, unlearn_config=unlearn_config, raw_datasets=  20
    raw_datasets, tokenizer=tokenizer)
                                                                               21
# Start unlearning and evaluation                                             22
trainer.unlearn()                                                              23
```

Listing 1: Example usage of MU-Bench: deleting 5% data from BERT-base trained on IMDB.

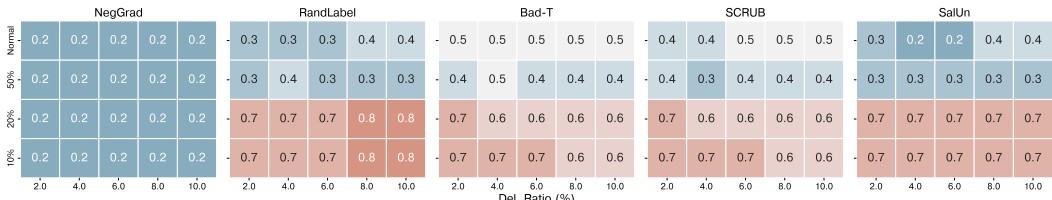

Figure 6: MU training with LoRA.

## A.4 Dataset level performance

We present the performance for each dataset in Figure 7-15.

## A.5 More results

We present the performance on LoRA in Figure 6, membership inference attack in Table 5 and unleanring time in Table 6.

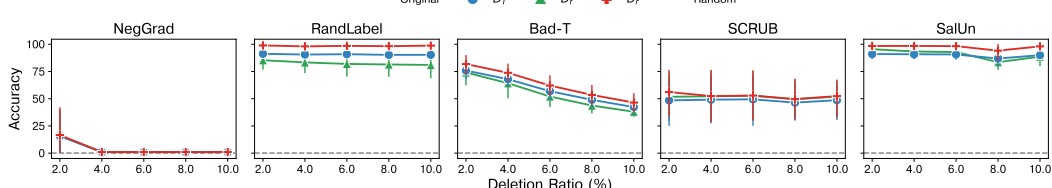

Figure 7: Performance on CIFAR-100.

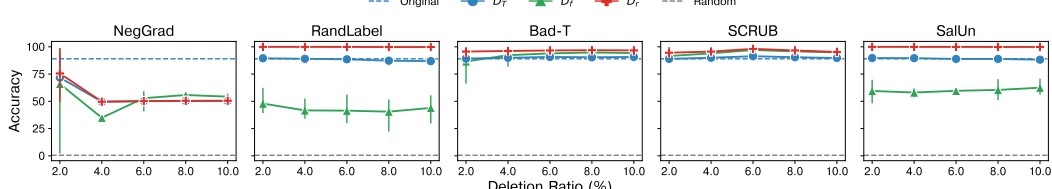

Figure 8: Performance on IMDB.

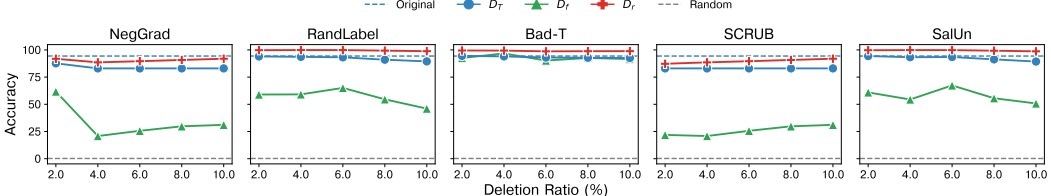

Figure 9: Performance on DDI-2013.

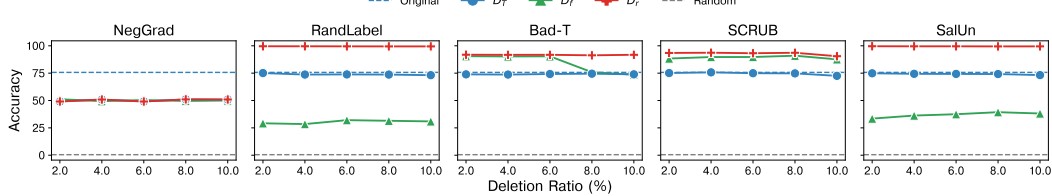

Figure 10: Performance on NLVR$^2$.

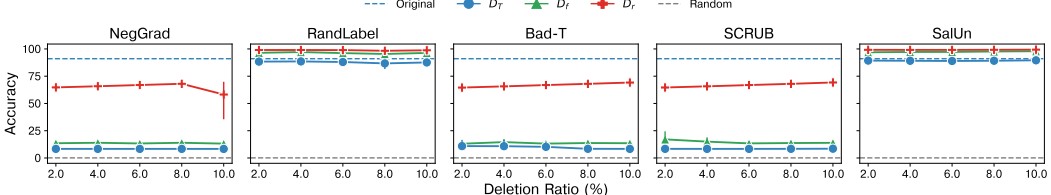

Figure 11: Performance on Speech Commands.

Table 4: Accuracy on $D_f$ for image generation task.

| Method | Acc ($\downarrow$) |
|---|---|
| NEGGRAD | 3.7 |
| RANDLABEL | 64.6 |
| BAD-T | 69.1 |
| SCRUB | 75.8 |
| SALUN | 48.2 |

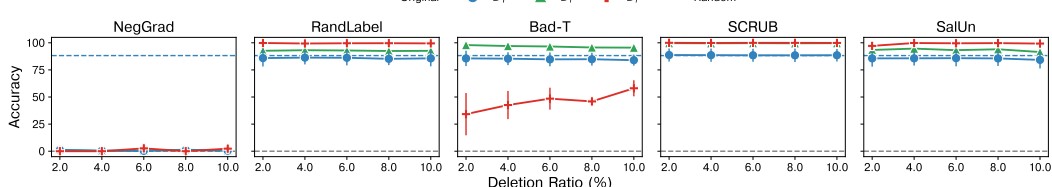

Figure 12: Performance on UCF101.

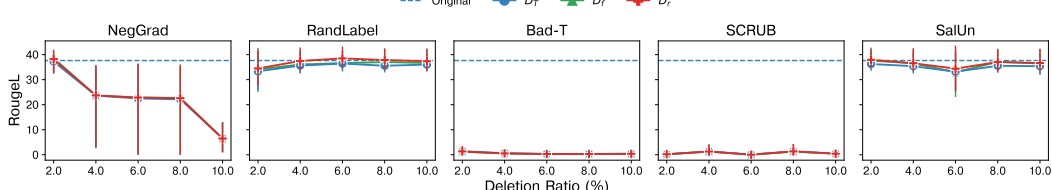

Figure 13: Performance on SAMSum.

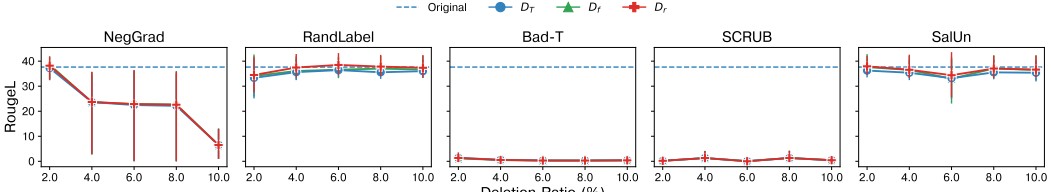

Figure 14: Performance on BioFact.

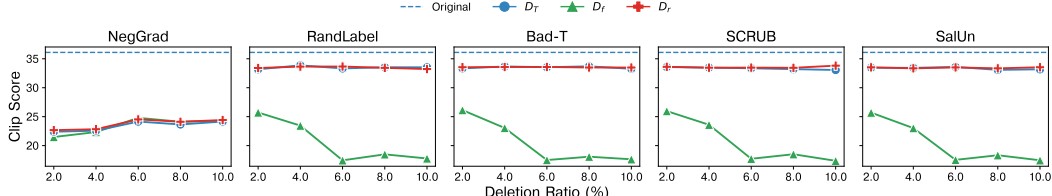

Figure 15: Performance on Tiny Imagenet.

Table 5: Success rate of membership inference attack.

| Method | Success Rate (%) ($\downarrow$) |
|---|---|
| NEGGRAD | 8.6 |
| RANDLABEL | 10.7 |
| BAD-T | 14.7 |
| SCRUB | 10.8 |
| SALUN | 11.5 |

Table 6: Average unlearning time across all datasets.

| Method | Unlearning time (hrs) ($\downarrow$) |
|---|---|
| NEGGRAD | 8.6 |
| RANDLABEL | 10.7 |
| BAD-T | 14.7 |
| SCRUB | 10.8 |
| SALUN | 11.5 |

