# OpenReview forum: "MU-Bench: A Multitask Multimodal Benchmark for Machine Unlearning"
_NeurIPS.cc/2024/Workshop/SafeGenAi — SafeGenAi Poster_

### Official Review · Reviewer_gyjH · 2024-10-09
**Good paper that introduces a machine unlearning benchmark with comprehensive experimental results evaluating the various algorithms**

**Rating:** 8
**Confidence:** 3

**Review:**

Summary: The authors introduce MU-Bench, a comprehensive machine unlearning benchmark that covers different models, modalities and tasks. The authors then evaluate different machine unlearning methods, and perform a detailed analysis across the MU-Bench.

Strength: the paper provides comprehensive and detailed experimental results, which can serve as fair comparisons among those different machine unlearning algorithms. Through the extensive experiments, the paper also addresses some of the interesting and important questions in the field, such as the scaling law of machine unlearning, the effect of curriculum learning and so on. The paper provides some new insights into the advantages and disadvantages of those different methods.

Weakness: 1. Many of the models used in this paper are not the most recent. I am wondering if there will be more models added to the package, which will include the newest open-sourced large language models. 2. Some important discussions, such as those on scaling laws and curriculum learning, are not linked to the relevant figures in the paper. Including these references would make the analysis clearer and easier to follow.

Overall, this is a solid paper that compares various machine unlearning algorithms across different models, data and modalities. The unified setup allows for a comprehensive understanding of these algorithms

---

### Official Review · Reviewer_mJBJ · 2024-10-09
**Review for "MU-Bench: A Multitask Multimodal Benchmark for Machine Unlearning"**

**Rating:** 6
**Confidence:** 3

**Review:**

The manuscript introduces a comprehensive benchmark system designed to evaluate existing machine unlearning techniques across various tasks and data modalities. This benchmark, MU-Bench, provides a standardized framework for comparing the effectiveness of different unlearning approaches in a consistent and fair manner, including tasks involving image, text, speech, and video data. Additionally, the unified evaluation metrics and the inclusion of underexplored areas like speech and video classification are innovative, setting a standard for future research in machine unlearning. While MU-Bench establishes a valuable framework for the systematic evaluation of machine unlearning techniques across diverse data modalities and tasks, the paper’s selective inclusion of existing MU methods represents a notable limitation. By not encompassing a broader array of unlearning algorithms, the benchmark may fail to fully capture the breadth of the current machine unlearning landscape. This oversight could potentially exclude algorithms that might excel in the benchmark's diverse scenarios or address unique challenges that the included methods do not.

---

### Official Review · Reviewer_aVSw · 2024-10-10
**MU-Bench - A Comprehensive Machine Unlearning Benchmark - Strong Paper**

**Rating:** 9
**Confidence:** 3

**Review:**

Review Overview:\
MU-Bench provides a structured machine unlearning benchmark. The benchmark is well-motivated and well-constructed. The paper presents comprehensive experiments on this benchmark with previous works and many additional experiments, providing insights into these machine unlearning methods. This paper is highly informative and would be a valuable contribution to the community.\
This review follows the sections of the paper. Strengths (+) and weaknesses (-) are noted for each section.

Introduction:\
\+ Good intro, brief and to the point. It seems the community could benefit from this benchmark.\
\+ The paragraph at the end of the intro summarizing results is strong.

MU-Bench:\
\+ Good diversity in domains, although it would be nice to see sensor modalities other than video/text.\
\- Impressive taxonomy of existing techniques but slightly hard to follow for someone unfamiliar with these methods. It might be helpful to take one method as an example and walk through each of its components.

Experiments:\
\+ Strong empirical results using 5 repeated trials with random seeds.\
\- It would be nice to plot the error bars or standard deviation in Figure 3, instead of just the averages.\
\- The main results figures should be moved from the appendix to the experiment section. It is very difficult to interpret the results described in this section while having to scroll down to page 22. To save space, you could try combining all the plots into one grid and use the labels and titles presented at the top only once. Alternatively, discuss only 2 or 3 datasets whose results fit, and in one or two sentences say "the methods succeed at forgetting in biomedical, but not in visual reasoning, speech, or action; refer to the appendix for more details" and move all the rest of these dataset discussions to the appendix.

Discussion and Analysis:\
\- If you have any insights or hypotheses on why Negrad affects bias, or why SCRUB doesn't scale, it would be beneficial to mention them.\
\+ Interesting results in curriculum learning, with lots of open opportunities for future work to build off of from this Discussion section.\
\- Lines 393-394: Low score on D_f might reveal information to an adversary. This is an interesting claim. Do you have any thoughts on how to test it? Or any example scenarios where this could be the case? Have any previous works mentioned this?

Conclusion:\
\+ Great summary.